Subject Areas:
ecology/environmental science

Keywords:
sea cows, marine mammals, extinct, South China Sea, local ecological knowledge

Author for correspondence:
Songhai Li
e-mail: lish@idsse.ac.cn

# Functional extinction of dugongs in China

Mingli Lin[1], Samuel T. Turvey[2], Chouting Han[1,3], Xiaoyu Huang[1,3], Antonios D. Mazaris[4], Mingming Liu[1], Heidi Ma[2], Zixin Yang[1], Xiaoming Tang[1,3] and Songhai Li[1,5]

[1]Marine Mammal and Marine Bioacoustics Laboratory, Institute of Deep-sea Science and Engineering, Chinese Academy of Sciences, Sanya 572000, People's Republic of China
[2]Institute of Zoology, Zoological Society of London, Regent's Park, London NW1 4RY, UK
[3]University of Chinese Academy of Sciences, Beijing 100049, People's Republic of China
[4]Department of Ecology, School of Biology, Aristotle University, UPB 119, Thessalonica 54124, Greece
[5]Center for Ocean Mega-Science, Chinese Academy of Sciences, Qingdao 266071, People's Republic of China

 STT, 0000-0002-3717-4800; SL, 0000-0003-4977-1722

Dugongs (*Dugong dugon*) experienced a serious population decline in China during the twentieth century, and their regional status is poorly understood. To determine their current distribution and status, we conducted a large-scale interview survey of marine resource users across four Chinese provinces and reviewed all available historical data covering the past distribution of dugongs in Chinese waters. Only 5% of 788 respondents reported past dugong sightings, with a mean last-sighting date of 23 years earlier, and only three respondents reported sightings from within the past 5 years. Historical records of dugongs peak around 1960 and then decrease rapidly from 1975 onwards; no records are documented after 2008, with no verified field observations after 2000. Based on these findings, we are forced to conclude that dugongs have experienced rapid population collapse during recent decades and are now functionally extinct in China. Our study provides evidence of a new regional loss of a charismatic marine megafaunal species, and the first reported functional extinction of a large vertebrate in Chinese marine waters. This rapid documented population collapse also serves as a sobering reminder that extinctions can occur before effective conservation actions are developed.

## 1. Introduction

A major challenge in conservation biology is to determine the dynamics of local extinctions across species' ranges, as progressive population losses and range contractions ultimately

lead to species extinctions [1]. Understanding the processes implicated in regional extinctions can improve conservation capacity and ability to protect surviving populations elsewhere [1]. Unfortunately, such information is not always available. This is especially the case for charismatic marine megafauna, which typically have broad geographical distributions, use various habitats over their long lifespans, and are thus potentially at risk from multiple anthropogenic threats [2].

Dugongs (*Dugong dugon*) are the only strictly marine herbivorous mammals, and the only extant species in the family Dugongidae [3]. They inhabit coastal waters of 37 tropical and subtropical countries from East Africa to Vanuatu, and as far north as the southwestern islands of Japan [4,5]. Dugongs are historically recorded from almost all the countries that border the South China Sea (SCS), including China, Vietnam, Cambodia, Thailand, Malaysia, Brunei and the Philippines [3]. They have been documented in Chinese waters for several hundred years [6] and into the twentieth century [7–9], when the species was mainly recorded in the coastal waters of Beibu Gulf, where shallow waters with abundant seagrass represented important habitat [10,11].

However, the near-shore habitats inhabited by dugongs overlap highly with the activity area of fishers and other marine resource users, making them vulnerable to human pressures. The species is listed as globally Vulnerable by IUCN [4], and as a Grade 1 National Key Protected Animal since 1988 by the Chinese State Council. Its recent status in Chinese waters is poorly understood; it is considered extinct in Taiwan, with sporadic stranding records suggesting continued survival elsewhere across this region, but with limited field surveys conducted to assess its distribution [3–5]. To address this conservation knowledge gap, we conducted a new interview survey to collect local ecological knowledge across the entire historical range of dugongs in mainland Chinese waters, and investigated dugong status and threats by analysing historical records from fishery captures, strandings, bycatch, field surveys and accidental sightings.

## 2. Material and methods

An interview survey was conducted in fishing communities along the coastal region of the northern SCS, including 66 villages in 22 municipalities across four southern Chinese maritime provinces (Hainan, Guangxi, Guangdong, Fujian). Our historical data review indicates that these sites cover the entire known range of dugongs in mainland Chinese waters (figure 1*a*) [11]. Spatial survey design was based upon information about number and location of fishing ports provided by the China Fishery Statistical Yearbook [12] and by local Ocean and Fishery bureaux. These sources also provided information on the number of registered fishing boats/families in each province and municipality, which was used to guide proportional sampling.

Interviews were conducted between 15 July and 13 August 2019, by four researchers with knowledge of marine mammals and 21 trained volunteers recruited from local universities (most of whom were studying marine biology or ecology). Volunteers received training in dugong identification and interview skills before conducting interviews. We only interviewed professional fishers (i.e. people who practised fishing as their main source of economic income) aged 18 or above, who had lived and worked in the survey area for at least 5 years. We did not otherwise use respondent age, sex, ethnicity or other demographic characteristics as selection criteria. Respondents were selected through random encounters in fishing communities [13–15] and interviewed in Mandarin or Cantonese on a one-to-one basis in relaxed informal settings. We only conducted interviews following verbal consent of participants and assured them that data would be kept anonymously, and that they could withdraw at any time or choose not to answer any questions.

Local ecological knowledge data were collected following a standardized methodological protocol developed for conservation research in Chinese fishing communities [16] and previously applied to research on SCS marine mammals [17,18]. Interviews were conducted using a questionnaire that took approximately 45 min to complete and included a combination of multiple choice, short free response and multi-part questions (electronic supplementary material). As part of a wider series of questions about the status of the SCS ecosystem and its fisheries, we asked respondents about their age, education and fishing experience (e.g. how many years ago they started fishing, how many days per year they typically spent fishing, fishing area), and their perception of the present status of the SCS ecosystem and its fisheries. We showed two photos and one illustration of a dugong, showing the animal from different angles, and asked respondents whether they could identify it and had ever seen it in the wild, together with information on timing, location and frequency of sightings. To ensure that

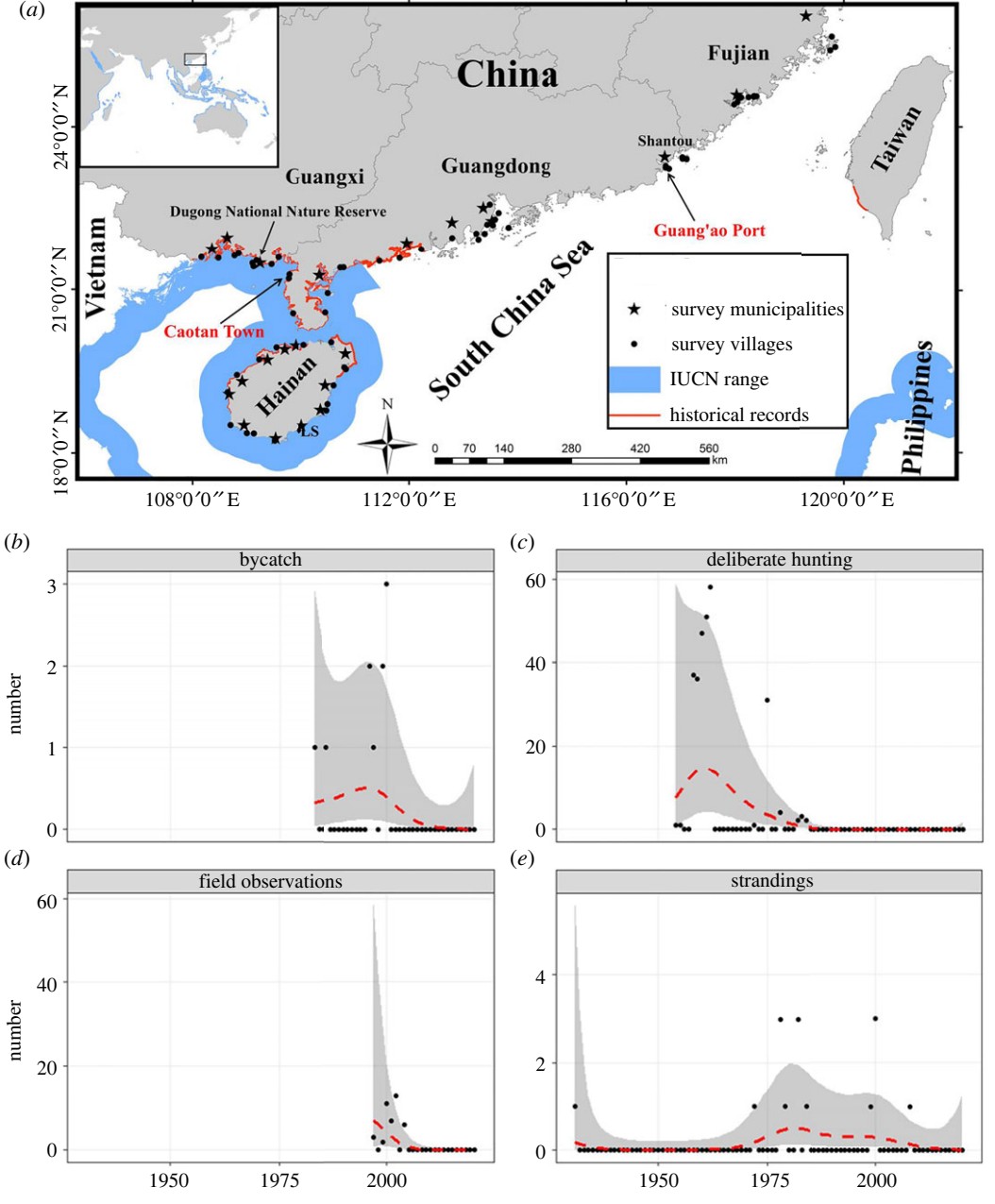

**Figure 1.** Distribution and population trend of dugongs (*Dugong dugon*) in China. (*a*) Distribution of dugongs and questionnaire survey locations in China and neighbouring waters. (*b–e*) GAMs showing effect of time on number of dugong historical records, based on different data sources (bycatch, deliberate hunting, field observations and strandings).

respondents could identify dugongs correctly, we asked them to name the species and describe how to distinguish it from other large marine vertebrates.

All available historical dugong records for China from the twentieth century onwards were collected (electronic supplementary material, table S1). We investigated trends in detected number of dugongs over time with generalized additive models (GAMs). As historical dugong records were collected through different data-generating processes that can vary over time, we treated different data sources (bycatch, deliberate hunting, field observations and strandings) as separate time series using a hierarchical smoother [19] to provide independent information on dugong numbers over time. We included years with a zero dugong count to avoid the fitted curve falling below zero. As count data have a strong mean–variance relationship that can affect estimated trends for small-count data [20] and do not meet assumptions of a Gaussian distribution, we used a negative binomial distribution to allow for clustering of observations. GAMs were fitted in the 'mgcv' library in R v. 4.1.1 (see electronic

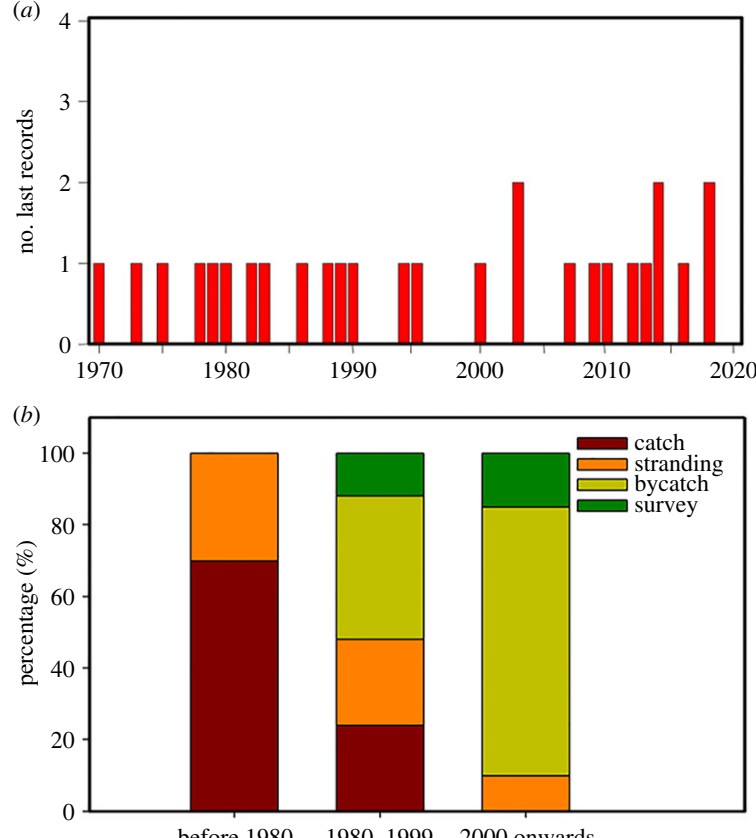

**Figure 2.** (*a*) Frequency distribution of 26 dugong last-sighting records from 1970 to 2019 reported by respondents in our interview survey (red bars); (*b*) changing sources of dugong observations across the late twentieth and early twenty-first centuries based on historical records.

supplementary material for R code) and built using the restricted maximum-likelihood method with $k = 10$, with a 'fs' smoother used for basis functions construction [21].

## 3. Results

We interviewed 788 respondents (electronic supplementary material, table S2). Mean respondent age was 51 years (range = 19–105; s.d. = 13). Most respondents were men (95.6%) and educated to junior middle-school level (87.9%). They had been fishers for a mean of 25 years (s.d. = 14), typically spent $185 \pm 67$ days per year fishing, and most (86.7%) fished within 100 km of the coast (mean = 66 km offshore). Our respondent sample therefore conducted regular fishing activities in near-shore marine habitats that represent primary dugong habitat, making them likely to be familiar with the regional status of the species.

In total, 12% of respondents ($N = 91$) recognized a picture of a dugong, but only 5% ($N = 37$) reported past sightings, with a mean last-sighting date of 23.2 years earlier and a median last-sighting date of 24.5 years earlier (s.d. = 16; figure 2$a$). Respondents in 13 municipalities (59%) reported dugong sightings, but fewer than five sightings were reported from each municipality except for Beihai, Guangxi Province, where China's Dugong National Nature Reserve is located and where 11 of 57 respondents (19%) reported sightings. Overall, only three respondents reported sightings from within the past 5 years, including one sighting near the dugong reserve in Caotan, western Zhanjiang Prefecture (western Guangdong), and two sightings in the Guang'ao Port Area of Shantou Prefecture (eastern Guangdong), an area with no historical dugong records. One bycatch event was reported (in 2000 at Beihai), and two historical stranding events were reported (at Beihai in 1983 and at Danzhou, Hainan in 1956).

Historical data analysis indicates that no published or unpublished records of dugongs are documented in China after 2008, with no verified field observations after 2000. According to

unpublished fisheries records, 257 dugongs were hunted for food between 1958 and 1976 (mean ± s.d. = 13.79 ± 4.89 per year), with only a few individuals reported after this period as incidental sightings, accidental bycatch and anecdotal fisher reports (electronic supplementary material, table S1). Before 1980, deliberate hunting and strandings were the main sources through which dugongs were detected, whereas after 1980 animals were mainly detected through bycatch and field observations (figure 2*b*). GAMs fitted to these data showed a strongly uneven temporal distribution pattern ($p < 0.001$, REML = 170.19, $R^2 = 0.302$, $N = 219$), and declining trends across time series were observed for all data sources (figure 1*b–d*), although stranding records showed a flatter decreasing trend (figure 1*e*).

## 4. Discussion

Based upon assessment of historical records, recent sightings and temporal trends, we present robust evidence revealing a rapid population collapse and potential functional extinction of dugongs in China. Very few respondents with extensive experience of Chinese coastal ecosystems have ever seen the species, with only three out of 788 respondents reporting sightings from the past 5 years, and there have been no confirmed records since 2008. We acknowledge the possibility that a few surviving dugongs in Chinese coastal waters might have been undetected by respondents. We also recognize the potential for species misidentification and incorrect recall of sighting locations and/or timings by local resource users, even those possessing considerable expertise on regional biodiversity, although recall errors are more likely with older sightings that are less informative for understanding recent dugong status [22]. It is therefore possible that a remnant dugong population might conceivably survive in the northern SCS, and we cannot confirm complete regional extirpation of the species.

However, our comprehensive assessment suggests that even if some individual dugongs still remain in Chinese waters, the dramatic population decline experienced by the species in recent decades is highly unlikely to be halted or reversed under current conditions, with the continuing deterioration of coastal ecosystems in the northern SCS meaning that dugongs have minimal hope of even short-term survival if they have not already disappeared. This negative assessment is strengthened by the fact that two of the three recent possible dugong sightings collected during our survey were from Shantou in eastern Guangdong Province, which is geographically far from the distribution of known historical dugong records around mainland China and lacks seagrass beds to support a permanent dugong population [23], but is relatively close (about 600 km) to the northern Philippines, a region that supports a separate dugong population (figure 1) [4,5]. Due to the known potential for long-distance movement by dugongs [24], these possible sightings might therefore represent vagrant individuals from the Philippine dugong population rather than animals from a surviving Chinese population, highlighting caveats with analysis of recent sighting reports that might not be representative of the target Chinese dugong population. While it is also possible that dugongs in Chinese waters might have shifted their distribution northward along the coastline to other habitat refugia in response to human activities or climate change across their original range, this possibility is unlikely as seagrass beds are largely degraded across the entire northern SCS [25], and no other dugong sightings or strandings have been reported from the Shantou area. We therefore conclude that there is no evidence to suggest that a remnant dugong population can persist within the northern SCS, and the species is likely to be functionally extinct (i.e. no longer able to maintain a viable population) in China.

While we welcome any possible future evidence that dugongs might survive in China, our study provides important evidence of the probable regional loss of a charismatic marine megafaunal species, and the first functional extinction of a large vertebrate in Chinese coastal waters. The probable disappearance of dugongs from China leaves the northern margin of the species' distribution with an isolated, threatened remnant population in southwestern Japan, and the closest surviving Asian population in northern Philippines and southern Vietnam [3–5,26]. Dugong extinction in China follows the recent extinction of the Yangtze River dolphin or baiji (*Lipotes vexillifer*) [27] and repeats the fate of the other historically occurring eastern Asian dugongid species, Steller's sea cow (*Hydrodamalis gigas*), which was hunted to extinction within 27 years of its discovery in the eighteenth century [28]. This rapid documented population collapse also serves as a sobering reminder that local extinction can happen within a very short time, especially for long-lived, late-maturing species with low reproductive rates, and potentially before effective conservation actions can be developed within dugong habitats in other countries. Deliberate hunting combined with the degradation of seagrass beds and accidental entanglement probably together contributed to the rapid collapse of China's dugong population [8,25], and the functional extinction of dugongs reflects the latest stage in the

progressive ecological deterioration of marine ecosystems in Chinese waters, which are home to approximately one-third of the world's marine mammal species [29,30]. Many of these other species are also experiencing rapid declines [31,32], emphasizing the urgent need to adopt more sustainable regional marine stewardship practices and optimize marine conservation efforts.

The IUCN already recognizes dugongs as extinct in Taiwan [4], and based upon our findings, we recommend that the species' wider regional status should be reassessed as Critically Endangered (Possibly Extinct) across the entirety of Chinese waters [33]. Further surveys should be conducted across neighbouring regions to confirm the geographical extent of dugong disappearance across the wider SCS. The message is alarming, reminding us that effective population and habitat management are critically needed within dugong habitats elsewhere. Conversely, although dugongs are probably now extirpated from China, efforts to evaluate, conserve and recover the region's seagrass ecosystem, a key habitat for dugongs and wider regional biodiversity, is a conservation priority in Chinese waters along with the protection of other marine habitats. Current rapid economic growth in China and other countries around the SCS, demonstrated through the recent increase in coastal economic activities (fishing, boat-based tourism, marine construction), is altering the structure and function of critical marine habitats and impacting marine mammals and wider biodiversity across the region. Our study highlights that immediate and extreme measures are necessary to prevent further extinctions of other keystone species in the wider SCS ecosystem. Improved monitoring is essential to identify species at high risk of extinction and guide regional conservation actions. The extinction of an emblematic species such as the dugong in China raises further concerns for other threatened marine mammals within a system where human activities now dominate the seascape.

Ethics. Fieldwork was approved by the Research Ethics Committee of the Institute of Deep-sea Science and Engineering, Chinese Academy of Sciences (IDSSE-SYLL-MMMBL-01).

Data accessibility. The data are published in the article and provided as electronic supplementary material [34].

Authors' contributions. M.L.: conceptualization, data curation, formal analysis, funding acquisition, investigation, methodology, project administration, software, supervision, visualization, writing—original draft and writing—review and editing; S.T.T.: conceptualization, formal analysis, methodology, supervision and writing—review and editing; C.H.: data curation, investigation and methodology; X.H.: data curation, investigation and methodology; A.D.M.: formal analysis, writing—original draft and writing—review and editing; M.L.: methodology and writing—review and editing; H.M.: methodology and writing—review and editing; Z.Y.: investigation and project administration; X.T.: data curation; S.L.: conceptualization, funding acquisition, project administration, resources, supervision and writing—review and editing.

All authors gave final approval for publication and agreed to be held accountable for the work performed therein.

Conflict of interest declaration. We declare we have no competing interests.

Funding. This research was financially supported by the Biodiversity Investigation, Observation and Assessment Programme (2019–2023) of the Chinese Ministry of Ecology and Environment, the incubating programme of the Institute of Deep-sea Science and Engineering, Chinese Academy of Sciences (grant no. Y960041001), Ocean Park Conservation Foundation Hong Kong (grant no. AW02-1920), the National Natural Science Foundation of China (grant no. 41422604), 'One Belt and One Road' Science and Technology Cooperation Special Programme of the International Partnership Programme of the Chinese Academy of Sciences (grant no. 183446KYSB20200016), and the Key Deployment Project of the Centre for Ocean Mega-Science of the Chinese Academy of Sciences (grant no. COMS2020Q15).

Acknowledgements. We are grateful to 21 volunteers for assisting with the questionnaire survey, to the Blue Ribbon Association for recruiting volunteers, and to the many fishing communities that we visited for sharing their knowledge.

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
