## [Peer Review File · Royal Society Open Science]

Review History

RSOS-211994.R0 (Original submission)

Review form: Reviewer 1

Is the manuscript scientifically sound in its present form?

No

Are the interpretations and conclusions justified by the results?

No

Is the language acceptable?

Yes

Do you have any ethical concerns with this paper?

No

Have you any concerns about statistical analyses in this paper?

Yes

Recommendation?

Major revision is needed (please make suggestions in comments)

Comments to the Author(s)

I have two main comments. First, I do not understand why the Authors refer to functional extinction. Functional extinction has a specific meaning. Specialized methods have been developed to detect it. I understand that the Authors' data are insufficient to support such analysis. But nothing in the analysis that they do perform points to functional extinction - as opposed, say, to a population collapse. Second, the Authors should do more with their statistical analysis. This is important because the paper is quite thin. Here's what I have in mind. Let $Y_j(t)$ be the sighting count by method j in year t . A natural statistical model is that $Y_j(t)$ has a Poisson (or negative binomial) distribution with mean $b_j * f(t)$, where b_j is a method effect and $f(t)$ is a population trend common to all methods. For this model to be identifiable, there is a need to put a constraint on the b_j 's, such as $b_1 = 1$ for a baseline method. The advantage of this approach is that allows the data for all methods to be used in estimating $f(t)$ - which is the quantity of interest - but accounting for differences between methods. It is straightforward to fit this model. Now, it is possible that the hierarchical GAM used by the Authors does this or something like it. If that's the case, then they need to spell it out carefully and focus on the common trend. (I rather doubt that their method actually does this because their Figure 1 appears to show different trend behavior for the different methods.)

Review form: Reviewer 2 (Eric Pedersen)

Is the manuscript scientifically sound in its present form?

No

Are the interpretations and conclusions justified by the results?

Yes

Is the language acceptable?

Yes

Do you have any ethical concerns with this paper?

No

Have you any concerns about statistical analyses in this paper?

No

Recommendation?

Accept with minor revision (please list in comments)

Comments to the Author(s)

I have reviewed this manuscript previously when it was submitted to Biology Letters. Overall, I thought the authors have effectively addressed mine and most of the other reviewer's comments, and I agree with the decision to remove the statistical model of extinction; currently, the manuscript relies on multiple lines of evidence to infer that the Dugong population is functionally extinct in Chinese waters, and a hypothesis test of functional extinction for one of the lines of evidence would not make the argument stronger than what is already presented.

I think the paper is presenting important information, and I agree that based on the data presented, a tentative conclusion of functional extinction in the region might be warranted. However, I would like to have a couple points clarified and expanded on.

First, I am not convinced that the authors fully addressed Reviewer 5's points from the last version of the paper. Reviewer 5 had requested additional information on movement and phylogeography of Dugong populations. The authors assert that the recent sightings in Caotan Town, Zhanjiang City, and Shantou City are in areas without historical records of dugong, but it is not clear from the presented data the observations were from reported ranges. They also assert (line 173) that these might represent iterant individuals from a Philippine population 600 km away. However, there is no biological information given on dugong populations to determine if that is a long distance for dugongs to travel. They also do not consider the possibility that a (maybe much reduced) dugong population has shifted in its range, rather than going functionally extinct. I would like to see more information on basic dugong biology presented, including maximum movement distances for the species, if known, as well as locations of other dugong populations relative to the Chinese population; how geographically distinct was this population? Are these other populations also declining / declined?

The authors' point that the most recent sightings (in the last 5 years) occurred outside the historical range for dugong would be more convincing if a map of the sightings from historical records and interviews was included. I would suggest replacing panel b of figure 2 (which currently does not provide any information not already in figure 1) with a map like Figure 1A, except showing the spatial distribution of historical records, and reported last sightings from surveys, maybe coloured by date. It would also be useful to have an illustration of where other populations are the other observations could have come from.

Second, I would have liked to see some discussion of other potential explanations for the lack of recent sightings: e.g. range shifts, or range contractions to more remote regions. Historical sightings of dugong in Figure 1A are all near what I would guess to be heavily populated areas. Are there any areas where the population range might have contracted to that would be less frequently surveyed? This is one place where having biological information on dugongs in the paper (e.g. foraging area or home range size) would be useful for the reader.

Third: I had not noted this the first time I went through this paper, but on reading it now, I noticed that the source of historical records was not made clear in the paper itself (only the supplemental material). Based on the supplement, all the historical data was gleaned from published papers, not directly from databases or monitoring program records. All the recorded field survey data came from a paper published in 2012, and there are no cited publications after that referring to subsequent field surveys. Similarly, the record of strandings is based on papers published on strandings. If there are any strandings that have occurred subsequently, or field monitoring sightings that have not been published, they would appear as zeros in this data. Are there reporting requirements for bycatch and for strandings? Are there any governmental or other bodies that engage in systematic record collection of this data? Is there any way of knowing whether bycatch or stranding occurred since the last published paper on the topic?

Minor comments:

Line 39: "mgcv" should be in lower case.

Line 109: "We included years with a zero dugong count to avoid the fitted curve falling below zero": this justification does not make sense. A negative binomial GAM with a log-link cannot give predictions below zero (i.e. have the curve falling below zero) in any circumstance. Including years without any observations as zeros instead avoids the issue of biasing abundances

or trends upward: if there are mostly observations of 1 or 2 individuals even when a population is at its most abundant, not including zero-observation years could mask a decline in population (as observations would just become less frequent, which would not cause the GAM curve to decline). Also, years without observations are informative; not seeing a dugong in a given year tells us something about the status of the population. However, this is predicated on these being true zeros (e.g. not just zero because bycatch or stranding was observed but had not yet been reported); see my third point above.

Line 127: Median last sighting date may be more relevant here than the mean. A mean can be quite sensitive to a few large values; the median sighting date directly tells the reader that 50% of the sightings occurred more than a specific time ago.

Line 173: Could this also not represent a range shift?

Figure 2: Given that you distinguish between observations within and outside the range of historical records, I would suggest color-coding the observations in Figure 2A as either occurring inside or outside the range.

Supplementary material: The information on the locations and dates of reported last sightings by fishermen is not currently available in the supplement. I understand that some of this needs to be kept private to maintain individual anonymity, it seems like some sort of summary table for the individuals who reported dugong sightings, and the locations of those sightings, could be provided.

Decision letter (RSOS-211994.R0)

Dear Dr Li

The Editors assigned to your paper RSOS-211994 "Functional extinction of dugongs in China" have now received comments from reviewers and would like you to revise the paper in accordance with the reviewer comments and any comments from the Editors. Please note this decision does not guarantee eventual acceptance.

Please submit your revised manuscript and required files (see below) no later than 21 days from today's (ie 31-Jan-2022) date. Note: the ScholarOne system will 'lock' if submission of the revision is attempted 21 or more days after the deadline. If you do not think you will be able to meet this deadline please contact the editorial office immediately.

on behalf of Prof Pete Smith (Subject Editor)
openscience@royalsociety.org

Associate Editor Comments to Author:

The reviewers highlight a number of concerns with your work - none are likely to require a great investment of time and effort, but they are important to tackle nevertheless. Note that reviewer 2 indicates that your dataset needs to be revisited (in that it doesn't currently appear to show an important aspect of the data). We'll look forward to receiving the revision in due course, and we will return the revised manuscript to the referees for their further consideration.

Reviewer comments to Author:

Reviewer: 1

Comments to the Author(s)

I have two main comments. First, I do not understand why the Authors refer to functional extinction. Functional extinction has a specific meaning. Specialized methods have been developed to detect it. I understand that the Authors' data are insufficient to support such analysis. But nothing in the analysis that they do perform points to functional extinction - as opposed, say, to a population collapse. Second, the Authors should do more with their statistical analysis. This is important because the paper is quite thin. Here's what I have in mind. Let $Y_j(t)$ be the sighting count by method j in year t . A natural statistical model is that $Y_j(t)$ has a Poisson (or negative binomial) distribution with mean $b_j * f(t)$, where b_j is a method effect and $f(t)$ is a population trend common to all methods. For this model to be identifiable, there is a need to put a constraint on the b_j 's, such as $b_1 = 1$ for a baseline method. The advantage of this approach is that allows the data for all methods to be used in estimating $f(t)$ - which is the quantity of interest - but accounting for differences between methods. It is straightforward to fit this model. Now, it is possible that the hierarchical GAM used by the Authors does this or something like it. If that's the case, then they need to spell it out carefully and focus on the common trend. (I rather doubt that their method actually does this because their Figure 1 appears to show different trend behavior for the different methods.)

Reviewer: 2

Comments to the Author(s)

I have reviewed this manuscript previously when it was submitted to Biology Letters. Overall, I thought the authors have effectively addressed mine and most of the other reviewer's comments, and I agree with the decision to remove the statistical model of extinction; currently,

the manuscript relies on multiple lines of evidence to infer that the Dugong population is functionally extinct in Chinese waters, and a hypothesis test of functional extinction for one of the lines of evidence would not make the argument stronger than what is already presented.

I think the paper is presenting important information, and I agree that based on the data presented, a tentative conclusion of functional extinction in the region might be warranted. However, I would like to have a couple points clarified and expanded on.

First, I am not convinced that the authors fully addressed Reviewer 5's points from the last version of the paper. Reviewer 5 had requested additional information on movement and phylogeography of Dugong populations. The authors assert that the recent sightings in Caotan Town, Zhanjiang City, and Shantou City are in areas without historical records of dugong, but it is not clear from the presented data the observations were from reported ranges. They also assert (line 173) that these might represent itinerant individuals from a Philippine population 600 km away. However, there is no biological information given on dugong populations to determine if that is a long distance for dugongs to travel. They also do not consider the possibility that a (maybe much reduced) dugong population has shifted in its range, rather than going functionally extinct. I would like to see more information on basic dugong biology presented, including maximum movement distances for the species, if known, as well as locations of other dugong populations relative to the Chinese population; how geographically distinct was this population? Are these other populations also declining / declined?

The authors' point that the most recent sightings (in the last 5 years) occurred outside the historical range for dugong would be more convincing if a map of the sightings from historical records and interviews was included. I would suggest replacing panel b of figure 2 (which currently does not provide any information not already in figure 1) with a map like Figure 1A, except showing the spatial distribution of historical records, and reported last sightings from surveys, maybe coloured by date. It would also be useful to have an illustration of where other populations are the other observations could have come from.

Second, I would have liked to see some discussion of other potential explanations for the lack of recent sightings: e.g. range shifts, or range contractions to more remote regions. Historical sightings of dugong in Figure 1A are all near what I would guess to be heavily populated areas. Are there any areas where the population range might have contracted to that would be less frequently surveyed? This is one place where having biological information on dugongs in the paper (e.g. foraging area or home range size) would be useful for the reader.

Third: I had not noted this the first time I went through this paper, but on reading it now, I noticed that the source of historical records was not made clear in the paper itself (only the supplemental material). Based on the supplement, all the historical data was gleaned from published papers, not directly from databases or monitoring program records. All the recorded field survey data came from a paper published in 2012, and there are no cited publications after that referring to subsequent field surveys. Similarly, the record of strandings is based on papers published on strandings. If there are any strandings that have occurred subsequently, or field monitoring sightings that have not been published, they would appear as zeros in this data. Are there reporting requirements for bycatch and for strandings? Are there any governmental or other bodies that engage in systematic record collection of this data? Is there any way of knowing whether bycatch or stranding occurred since the last published paper on the topic?

Minor comments:

Line 39: "mgcv" should be in lower case.

Line 109: “We included years with a zero dugong count to avoid the fitted curve falling below zero”: this justification does not make sense. A negative binomial GAM with a log-link cannot give predictions below zero (i.e. have the curve falling below zero) in any circumstance. Including years without any observations as zeros instead avoids the issue of biasing abundances or trends upward: if there are mostly observations of 1 or 2 individuals even when a population is at its most abundant, not including zero-observation years could mask a decline in population (as observations would just become less frequent, which would not cause the GAM curve to decline). Also, years without observations are informative; not seeing a dugong in a given year tells us something about the status of the population. However, this is predicated on these being true zeros (e.g. not just zero because bycatch or stranding was observed but had not yet been reported); see my third point above.

Line 127: Median last sighting date may be more relevant here than the mean. A mean can be quite sensitive to a few large values; the median sighting date directly tells the reader that 50% of the sightings occurred more than a specific time ago.

Line 173: Could this also not represent a range shift?

Figure 2: Given that you distinguish between observations within and outside the range of historical records, I would suggest color-coding the observations in Figure 2A as either occurring inside or outside the range.

Supplementary material: The information on the locations and dates of reported last sightings by fishermen is not currently available in the supplement. I understand that some of this needs to be kept private to maintain individual anonymity, it seems like some sort of summary table for the individuals who reported dugong sightings, and the locations of those sightings, could be provided.

===PREPARING YOUR MANUSCRIPT===

Your revised paper should include the changes requested by the referees and Editors of your manuscript. You should provide two versions of this manuscript and both versions must be provided in an editable format:
 one version identifying all the changes that have been made (for instance, in coloured highlight, in bold text, or tracked changes);
 a 'clean' version of the new manuscript that incorporates the changes made, but does not highlight them. This version will be used for typesetting if your manuscript is accepted.

If you have been asked to revise the written English in your submission as a condition of publication, you must do so, and you are expected to provide evidence that you have received

language editing support. The journal would prefer that you use a professional language editing service and provide a certificate of editing, but a signed letter from a colleague who is a fluent speaker of English is acceptable. Note the journal has arranged a number of discounts for authors using professional language editing services (<https://royalsociety.org/journals/authors/benefits/language-editing/>).

===PREPARING YOUR REVISION IN SCHOLARONE===

<https://royalsociety.org/journals/authors/author-guidelines/#supplementary-material> to

include a suitable title and informative caption. An example of appropriate titling and captioning may be found at https://figshare.com/articles/Table_S2_from_Is_there_a_trade-off_between_peak_performance_and_performance_breadth_across_temperatures_for_aerobic_sc_ope_in_teleost_fishes_/3843624.

Author's Response to Decision Letter for (RSOS-211994.R0)

See Appendix A.

RSOS-211994.R1 (Revision)

Review form: Reviewer 1

Is the manuscript scientifically sound in its present form?

No

Are the interpretations and conclusions justified by the results?

No

Is the language acceptable?

Yes

Do you have any ethical concerns with this paper?

No

Have you any concerns about statistical analyses in this paper?

Yes

Recommendation?

Reject

Comments to the Author(s)

I do not find the Authors' justification for referring to functional extinction compelling. The Authors appear not to have understood my suggestion for deepening their statistical analysis.

Review form: Reviewer 3 (Ellen Hines)

Is the manuscript scientifically sound in its present form?

Yes

Are the interpretations and conclusions justified by the results?

Yes

Is the language acceptable?

Yes

Do you have any ethical concerns with this paper?

No

Have you any concerns about statistical analyses in this paper?

No

Recommendation?

Accept with minor revision (please list in comments)

Comments to the Author(s)

Unfortunately I believe the sad conclusions here. What i would like to see however is more wording about the context and meaning of this extirpation to the range of the dugong..what does this mean both in China and nearby countries; internationally? IN MCGowen et al (2021), we found 2 dugong skulls in central Vietnam..certainly no dugongs there for many years..Taiwan? what is the context added to iucn assessments? another marine mammal gone from China? how will this affect current and future legislation? management actions? more about what this really means..glad to take another look.

Decision letter (RSOS-211994.R1)

Dear Dr Li

On behalf of the Editors, we are pleased to inform you that your Manuscript RSOS-211994.R1 "Functional extinction of dugongs in China" has been accepted for publication in Royal Society Open Science subject to minor revision in accordance with the referees' reports. Please find the referees' comments along with any feedback from the Editors below my signature.

Please submit your revised manuscript and required files (see below) no later than 7 days from today's (ie 18-Jul-2022) date. Note: the ScholarOne system will 'lock' if submission of the revision is attempted 7 or more days after the deadline. If you do not think you will be able to meet this deadline please contact the editorial office immediately.

on behalf of Prof Pete Smith (Subject Editor)
openscience@royalsociety.org

Associate Editor Comments to Author:

One of the original reviewers has provided further commentary on your paper, but recommends rejection, while a second (new) reviewer has provided other comments but recommends acceptance. This presents a challenge for the Editors, as the journal's policy is generally to reject outright if reviewers are not satisfied by the paper after major revision, but several of the reviewers (including those at Biology Letters) have now recommended acceptance after minor revision.

With the above in mind, but also recognising that you have provided extensive explanations and rebuttals for your decisions to amend (or not) your manuscript in particular ways, and the extensive review the paper has been subject to, the Editors are taking the view that it would be more useful to the community at large for your work to be available for discussion and comment in the literature, than to be rejected at this stage. The paper may, as a result, not be entirely 'correct', and the Editors recognise this, but the Editors' role is partly to adjudicate and make a judgement as to whether research as a whole is better served by some stimulating debate through useful but occasionally controversial papers than opting for 'safe' publications - and indeed, RSOS has a mission to publish good science, regardless of its impact.

We would nevertheless like you to do what you can to address the remaining comments in a final revision and also in another, equally comprehensive rebuttal, please.

Reviewer comments to Author:

Reviewer: 1

Comments to the Author(s)

I do not find the Authors' justification for referring to functional extinction compelling. The Authors appear not to have understood my suggestion for deepening their statistical analysis.

Reviewer: 3

Comments to the Author(s)

Unfortunately I believe the sad conclusions here. What i would like to see however is more wording about the context and meaning of this extirpation to the range of the dugong..what does this mean both in China and nearby countries; internationally? IN MCGowen et al (2021), we found 2 dugong skulls in central Vietnam..certainly no dugongs there for many years..Taiwan? what is the context added to iucn assessments? another marine mammal gone from China? how will this affect current and future legislation? management actions? more about what this really means..glad to take another look.

===PREPARING YOUR MANUSCRIPT===

one version should clearly identify all the changes that have been made (for instance, in coloured highlight, in bold text, or tracked changes);

===PREPARING YOUR REVISION IN SCHOLARONE===

- An individual file of each figure (EPS or print-quality PDF preferred [either format should be produced directly from original creation package], or original software format).
- An editable file of each table (.doc, .docx, .xls, .xlsx, or .csv).
- An editable file of all figure and table captions.

- Any electronic supplementary material (ESM).
- If you are requesting a discretionary waiver for the article processing charge, the waiver form must be included at this step.
- If you are providing image files for potential cover images, please upload these at this step, and inform the editorial office you have done so. You must hold the copyright to any image provided.
- A copy of your point-by-point response to referees and Editors. This will expedite the preparation of your proof.

- Ensure that your data access statement meets the requirements at <https://royalsociety.org/journals/authors/author-guidelines/#data>. You should ensure that you cite the dataset in your reference list. If you have deposited data etc in the Dryad repository, please only include the 'For publication' link at this stage. You should remove the 'For review' link.
- If you are requesting an article processing charge waiver, you must select the relevant waiver option (if requesting a discretionary waiver, the form should have been uploaded, see 'File upload' above).
- If you have uploaded any electronic supplementary (ESM) files, please ensure you follow the guidance at <https://royalsociety.org/journals/authors/author-guidelines/#supplementary-material> to include a suitable title and informative caption. An example of appropriate titling and captioning may be found at https://figshare.com/articles/Table_S2_from_Is_there_a_trade-off_between_peak_performance_and_performance_breadth_across_temperatures_for_aerobic_scope_in_teleost_fishes_/3843624.

Author's Response to Decision Letter for (RSOS-211994.R1)

See Appendix B.

Decision letter (RSOS-211994.R2)

Dear Dr Li:

I am pleased to inform you that your manuscript entitled "Functional extinction of dugongs in China" is now accepted for publication in Royal Society Open Science.

If you have not already done so, please ensure that you send to the editorial office an editable version of your accepted manuscript, and individual files for each figure and table included in

your manuscript. You can send these in a zip folder if more convenient. Failure to provide these files may delay the processing of your proof.

Please remember to make any data sets or code libraries 'live' prior to publication, and update any links as needed when you receive a proof to check - for instance, from a private 'for review' URL to a publicly accessible 'for publication' URL. It is also good practice to add data sets, code and other digital materials to your reference list.

Royal Society Open Science is a fully open access journal. A payment may be due before your article is published. Our partner Copyright Clearance Center's RightsLink for Scientific Communications will contact the corresponding author about your open access options from the email domain @copyright.com (if you have any queries regarding fees, please see <https://royalsocietypublishing.org/rsos/charges> or contact authorfees@royalsociety.org).

on behalf of Professor Pete Smith (Subject Editor).

Follow Royal Society Publishing on Twitter: @RSocPublishing
Follow Royal Society Publishing on Facebook:
<https://www.facebook.com/RoyalSocietyPublishing/>
Read Royal Society Publishing's blog:
<https://royalsociety.org/blog/blogsearchpage/?category=Publishing>

Sanya, 04/03/2022

Dear Editor,

We have now carefully considered all the comments made by the two reviewers, and have modified the manuscript accordingly. We believe that this revision has met the requirements for publication, and our paper has now been carefully reviewed by **nearly 10 reviewers in four rounds** since first submitted to *Biology Letters*. Thus, we would like to resubmit this improved version of the manuscript (RSOS-211994) to *Royal Society Open Science*.

We are grateful to the reviewers for their time in reviewing this MS and their helpful comments.

Best wishes,
Authors

Responses to the comments of Reviewer 1:

1. I have two main comments. First, I do not understand why the Authors refer to functional extinction. Functional extinction has a specific meaning. Specialized methods have been developed to detect it. I understand that the Authors' data are insufficient to support such analysis. But nothing in the analysis that they do perform points to functional extinction - as opposed, say, to a population collapse. Second, the Authors should do more with their statistical analysis. This is important because the paper is quite thin. Here's what I have in mind. Let $Y_j(t)$ be the sighting count by method j in year t . A natural statistical model is that $Y_j(t)$ has a Poisson (or negative binomial) distribution with mean $b_j * f(t)$, where b_j is a method effect and $f(t)$ is a population trend common to all methods. For this model to be identifiable, there is a need to put a constraint on the b_j 's, such as $b_1 = 1$ for a baseline method. The advantage of this approach is that allows the data for all methods to be used in estimating $f(t)$ - which is the quantity of interest - but accounting for differences between methods. It is straightforward to fit this model. Now, it is possible that the hierarchical GAM used by the Authors does this or something like it. If that's the case, then they need to spell it out carefully and focus on the common trend. (I rather doubt that their method actually does this because their Figure 1 appears to show different trend behavior for the different methods.)

➤ **Response:** Thank you very much for your time and constructive comments again. Your comments are very constructive, but we do have our own reservations. First of all, “functionally extinct” is a recognised ecological term that has been used in many other studies, for which we provide an accurate description in the text, but

there is no mathematic model which may help to make the conclusion that a species is functionally extinct, due to the inherent uncertainty around the status and continued survival of such species – indeed, it is this uncertainty which is captured by the term, and which makes it appropriate to use in this instance. We agree that it would be better if our conclusion could be made based on testing the null hypothesis that functional extinction of dugong has not yet occurred in China, and then calculating a p value to accept/reject this hypothesis. However, we don't agree that this is the only way to make this conclusion. For endangered marine mammals, it is always challenging or almost impossible to obtain reproductive or recruitment data. Does that mean these species will not go through the “functionally extinct” process? The baiji (*Lipotes vexillifer*) has been considered “functionally extinct” for many years, and this is widely accepted by the academic community, but this conclusion never been tested by the null hypothesis that functional extinction of baiji has not yet occurred. Moreover, the dugong populations of the Maldives, Lakshadweep Islands, Mauritius, Rodrigues, and Taiwan have all already been considered as extinct without a hypothesis testing approach.

Our data indicate that there is no evidence to suggest that a remnant dugong population can persist within this region, and so the species can be regarded as functionally extinct - i.e., it is no longer able to maintain a viable population. This conclusion was made based on “no records are documented after 2008” and “large-scale interview survey of marine resource users covering the past distribution of dugongs in China”. Of course, we can be more conservative, using phrases like “population collapse” or “severe decline”. Undoubtedly, these conservative phrases are riskless (and help us to mitigate pressure from government), but they can't well reflect the true situation and are not helpful to rouse people's concern about increasing environmental degradation in Chinese waters and the need to take urgent actions to protect marine biodiversity. We also feel strongly that such alternative wordings do not go far enough, based on our survey data – many species have declined or experienced population collapses, but are undoubtedly still in existence; whereas in this case, we are uncertain whether dugongs even still survive in China at all, and if they do, their uncertain status in the light of ongoing environmental degradation and other human pressures means that there is no hope for them, as they cannot even be found in the wild to conserve.

Thank you for the helpful comment about integrating the data from different methods to estimate a common population trend. Unfortunately, we can't do that in this case. Due to the incomparability of the data collecting methods, we can't set a baseline for b1. The unique thing that we can do is a PDP plot (partial dependency plot) to show the trend of variables in the model. Actually, building

the hierarchical GAM for each method was strongly suggested by another reviewer who is an expert in this field, and was specially invited by the editor of Biology Letters to inspect only assess use of the GAM model in this paper.

Responses to the comments of Reviewer 2:

1. I have reviewed this manuscript previously when it was submitted to Biology Letters. Overall, I thought the authors have effectively addressed mine and most of the other reviewer's comments, and I agree with the decision to remove the statistical model of extinction; currently, the manuscript relies on multiple lines of evidence to infer that the Dugong population is functionally extinct in Chinese waters, and a hypothesis test of functional extinction for one of the lines of evidence would not make the argument stronger than what is already presented. I think the paper is presenting important information, and I agree that based on the data presented, a tentative conclusion of functional extinction in the region might be warranted. However, I would like to have a couple points clarified and expanded on.

➤ **Response:** Thank you very much in agreement the publication of this MS again.

2. First, I am not convinced that the authors fully addressed Reviewer 5's points from the last version of the paper. Reviewer 5 had requested additional information on movement and phylogeography of Dugong populations. The authors assert that the recent sightings in Caotan Town, Zhanjiang City, and Shantou City are in areas without historical records of dugong, but it is not clear from the presented data the observations were from reported ranges. They also assert (line 173) that these might represent iterant individuals from a Philippine population 600 km away. However, there is no biological information given on dugong populations to determine if that is a long distance for dugongs to travel. They also do not consider the possibility that a (maybe much reduced) dugong population has shifted in its range, rather than going functionally extinct. I would like to see more information on basic dugong biology presented, including maximum movement distances for the species, if known, as well as locations of other dugong populations relative to the Chinese population; how geographically distinct was this population? Are these other populations also declining/declined?

➤ **Response:** Thank you for the comments. It is a good idea to add more information about the status of dugongs that are known to exist geographically near to the Chinese population. Unfortunately, except for the well-studied population in Australia, there are often only stranding records and anecdotal evidences available in many parts of the species' range. The information relative to this study can be summarized as follows:

As shown in Figure 1, there are three geographically proximal dugong populations possibly relevant to analysis of the Chinese population, which are

located in Japan, Vietnam, and the Philippines respectively (other dugong populations in the SCS, i.e. in Cambodia and Brunei, are very far from mainland China). The presence of dugongs within the southwestern Islands of **Japan** (200 km from Taketomi Island to Taiwan Province, and 450 km to Mainland China) has been well established for many centuries. However, little is known about the distribution of dugongs in this region. It has been concluded that this dugong population is extinct or in extremely low numbers on the basis that there were no sightings during an aerial survey in 1999; no feeding trails were observed on seagrass beds during the subsurface surveys. Historically, almost all the islands of the **Philippines** have recorded sightings of dugongs. Unfortunately, very limited ecological information is available. Undoubtedly there is a sizable population distributed in northern Philippines (about 600 km to Guangdong) based on frequent stranding reports. There have also been several reports of dugongs in **Vietnamese waters** since the 1960s, but there have been no recent surveys. There are unconfirmed reports of a sizable population of dugongs in southwest Vietnam, along the Gulf of Thailand coast in the vicinity of Phu Quoc Island (1500 km to Hainan). As such, we consider it most likely that of these three populations, the Philippine population might be most likely to constitute a source population for possible recent Chinese dugong sightings, as we discuss in our paper.

All dugong populations around the world are rapidly decreasing (including Australia which supports the largest populations of dugongs, e.g. there are 72,000 Dugongs in the early 1960s in urban coast of Queensland, compared with an estimated 4,220 in the mid 1990s), and several island populations including the Maldives, the Lakshadweep Islands, Mauritius, Rodrigues, and Taiwan are now considered extinct (IUCN 2019). There is also anecdotal evidence that the area of occupancy of the dugong has declined in many parts of its range, but specific population trends in Vietnam and Philippines are unclear due to lack of investigation. Based on IUCN reports, the outlook for dugongs in Vietnam, Japan, Cambodia and Brunei must be poor as these countries have very small dugong populations.

Dugongs show great variability in movement patterns and migration, depending on the study area and the influence of seasonal temperature or rainfall on regional ecosystems. Long-distance movements by dugongs along the Queensland coast are well-documented of up to 560 km, based on 14 individuals that travelled >100 km, with mean±S.E. macro-scale movement distance per individual of 243.8±35.4 km (Sheppard *et al.* 2006).

Including this level of detail is beyond the scope of this paper, but we have added extra text to summarize this information (lines 43-48 and lines 179-190). We have also now cited Sheppard *et al.* 2006, which supports large-scale movement

patterns and migration of dugongs (new reference 23).

Figure 1. The IUCN Red List of *Dugong dugon* – published in 2019.

3. The authors' point that the most recent sightings (in the last 5 years) occurred outside the historical range for dugong would be more convincing if a map of the sightings from historical records and interviews was included. I would suggest replacing panel b of figure 2 (which currently does not provide any information not already in figure 1) with a map like Figure 1A, except showing the spatial distribution of historical records, and reported last sightings from surveys, maybe coloured by date. It would also be useful to have an illustration of where other populations are the other observations could have come from.
➤ **Response:** It is a good suggestion to add a map showing the sighting locations from historical records and interviews. However, in addition to stranding sites, other sighting, hunting or bycatch locations reported by fishermen can't use for mapping due to great uncertainties of their fishing range (mean distance is 66 km to the port). According to your suggestion, we have now highlighted the locations (in Shantou and Beihai city) with the most recent dugong sightings reported by our informants in Figure 1.
4. Second, I would have liked to see some discussion of other potential explanations for the lack of recent sightings: e.g. range shifts, or range contractions to more remote regions. Historical sightings of dugong in Figure 1A are all near what I would guess to be heavily populated areas. Are there any areas where the population range might have contracted to that would be less frequently surveyed? This is one place where having biological information on dugongs in the paper

(e.g. foraging area or home range size) would be useful for the reader.

➤ **Response:** Thank you for the helpful comment. We understand the concerns about changes in dugong distribution, rather than the collapse of the population. This is why we need to use a large-scale fisherman questionnaire and collect all the historical present data to completely cover the entire historical recorded range of dugongs in China, to assess whether range shifts versus range contractions of dugongs can be distinguished. In addition, another reason to rule out range shift is data from recent seagrass surveys. Dugongs are obligate herbivores and their occurrence is closely related to the distribution of seagrass. Recent seagrass survey in the northern SCS showed that the area and density of seagrass continue to decrease (Figure 2 below), with only 74 km² remaining (56 km², with 76% of these areas located in eastern Hainan Island). It is thus impossible to support a sustainable population of dugong without large areas of seagrass meadow, which effectively no longer exist in Chinese waters.

We have now added extra text to discuss these other potential explanations for the lack of recent sightings: e.g. range shifts, or range contractions to more remote regions (Lines 185-190).

Figure 2: Geographic distribution of seagrass beds in South China. 1, Dapeng Bay seagrass bed; 2, Shenzhen Bay seagrass bed; 3, Hailing Island seagrassbed; 4, Donghai Island seagrass bed; 5, Liusha Bay seagrass bed; 6, Hepu seagrass bed; 7, Pearl Bay seagrass bed; 8, Longwan Bay seagrass bed; 9, Li'an Bay seagrass bed ; 10, Xincun Bay eagrass bed ; 11, Sanya Bay seagrass bed. (Cited from Huang et

al., 2006; Main seagrass beds and threats to their habitats in the coastal sea of South China).

5. Third: I had not noted this the first time I went through this paper, but on reading it now, I noticed that the source of historical records was not made clear in the paper itself (only the supplemental material). Based on the supplement, all the historical data was gleaned from published papers, not directly from databases or monitoring program records. All the recorded field survey data came from a paper published in 2012, and there are no cited publications after that referring to subsequent field surveys. Similarly, the record of strandings is based on papers published on strandings. If there are any strandings that have occurred subsequently, or field monitoring sightings that have not been published, they would appear as zeros in this data. Are there reporting requirements for bycatch and for strandings? Are there any governmental or other bodies that engage in systematic record collection of this data? Is there any way of knowing whether bycatch or stranding occurred since the last published paper on the topic?

➤ **Response:** This is a functionally extinct species, for which there are unfortunately no other recent records; indeed, it would be breaking news if even one individual had been discovered recently in China (It's like finding a baiji in the Yangtze River). We can only collect the historical data from published papers, as there is no record from databases or monitoring programmes (The stranding rescue network and monitoring programme for marine mammals in some provinces have been built, e.g. Hainan: <http://www.cetacean.csdb.cn/>, but these contain no dugong data). The data are mainly from a review published in 2012, as there is no record from after that (field surveys, stranding, bycatch or any other ways).

6. Line 39: “mgcv” should be in lower case.

➤ **Response:** Amend accordingly.

7. Line 109: “We included years with a zero dugong count to avoid the fitted curve falling below zero” : this justification does not make sense. A negative binomial GAM with a log-link cannot give predictions below zero (i.e. have the curve falling below zero) in any circumstance. Including years without any observations as zeros instead avoids the issue of biasing abundances or trends upward: if there are mostly observations of 1 or 2 individuals even when a population is at its most abundant, not including zero-observation years could mask a decline in population (as observations would just become less frequent, which would not cause the GAM curve to decline). Also, years without observations are informative; not seeing a dugong in a given year tells us something about the status of the population. However, this is predicated on these being true zeros (e.g. not just zero because bycatch or stranding was observed but had not yet been

reported); see my third point above.

- **Response:** “included years with a zero dugong count to avoid the fitted curve falling below zero” is suggested by another reviewer (an expert in this field, specially invited by the editor of *Biology Letters* to inspect the GAM model). A negative binomial GAM with a log-link can give predictions below zero as showed in following figure (You can build the model based on our data in Electronic supplementary material). There is a misunderstanding about “zero-observation years” here. There are no zero-observation years in our study, even if the observed effort may change with time. The efforts of stranding (population increased and social media evolved), bycatch (fishing effort rapidly increase) and field survey (more and more scientific surveys were conducted) increased during the study time, but hunting activity stopped. Thus, there is “zero dugong count” but not “zero-observation”.

8. Line 127: Median last sighting date may be more relevant here than the mean. A mean can be quite sensitive to a few large values; the median sighting date directly tells the reader that 50% of the sightings occurred more than a specific time ago.
 - **Response:** Thank you. For the last sighting date, both mean and median can be well describe the central tendency of data as no extreme value here. Mean last-sighting date of 23.23 years compared to median last sighting date of 24.5 year. We have now added both values to the Results section (Lines 131).
9. Line 173: Could this also not represent a range shift?
 - **Response:** We don't think this represent a range shift. Firstly, the percentage of fishermen reporting sightings was low, with only 2 out of 67 informers. Secondly,

except for this survey, there is a lack of any other evidence to support the presence of dugongs in this area (nor stranding, bycatch or incident sighting was available). As fishermen operate in areas up to 100 km, there is considerable uncertainty about the exact location of these two reports. Thirdly, no seagrass meadow is located in Shantou (above Figure 2).

10. Figure 2: Given that you distinguish between observations within and outside the range of historical records, I would suggest color-coding the observations in Figure 2A as either occurring inside or outside the range.
 - **Response:** Only 27 informants provided last-sighting dates, and only 10 of these dated from the past 20 years. Therefore, color-coding the observations as either occurring inside or outside the range seems not so important for considering the range shift, as the dataset is small and will have limited power to indicate any spatial patterns or trends. Moreover, all four surveyed provinces are considered to be part of the historical range of dugongs in China; although Fujian Province didn't have any historical records, it is near to Taiwan which is known to have formerly supported a local dugong population.

11. Supplementary material: The information on the locations and dates of reported last sightings by fishermen is not currently available in the supplement. I understand that some of this needs to be kept private to maintain individual anonymity, it seems like some sort of summary table for the individuals who reported dugong sightings, and the locations of those sightings, could be provided.
 - **Response:** All the information and data in the MS can be opened and no need to keep private. The fishermen can only have a vague impression about the sighting information, thus it is impossible to obtain the precise locations and date of those sightings. The information of most last sightings reported by fishermen has been presented directly in the text (Lines 135-141).

Sanya, 07/27/2022

Dear Editor,

We have now carefully considered the comments made by the Associate Editor and two reviewers, and have modified the manuscript accordingly. Thus, we would like to resubmit this improved version of the manuscript (RSOS-211994.R1) to *Royal Society Open Science*.

We are grateful to the reviewers for their time in reviewing this MS and helpful comments.

Best wishes,
Authors

Responses to the comments of Reviewer 1:

1. I do not find the Authors' justification for referring to functional extinction compelling. The Authors appear not to have understood my suggestion for deepening their statistical analysis.
➤ **Response:** We would like to thank the referee for all comments and suggestions as they offered us the opportunity to go deeper into the literature and provide all justifications on issues raised. In the last version of the manuscript we improved our statistical analyses based on the data that we had available. We hope that this revised version satisfies the referee.

Responses to the comments of Reviewer 2:

2. What i would like to see however is more wording about the context and meaning of this extirpation to the range of the dugong. What does this mean both in China and nearby countries; internationally? IN MCGowen et al (2021), we found 2 dugong skulls in central Vietnam. Certainly no dugongs there for many years.Taiwan? what is the context added to IUCN assessments? Another marine mammal gone from China? How will this affect current and future legislation? management actions? More about what this really means. Glad to take another look.
➤ **Response:** In the revised manuscript we make a clear case that this study provides new evidence about the status of dugongs in Chinese waters. Based on this result, in the final paragraph of the revised manuscript's discussion we now suggest that dugongs in Chinese waters should now be assessed as Critically Endangered (Possibly Extinct) by the IUCN (and we include a new reference to provide further information on this particular status category). We have also now added the new citation proposed by the referee, and highlight the need for further

dugong surveys in neighbouring regions of the South China Sea (SCS) to establish the geographic extent of dugong population loss across this wider region. This alarming message suggests that all necessary measures are critically needed to prevent further extinctions of other keystone species in the progressive ecological deterioration of the SCS ecosystems, which are home to approximately one-third of the world's marine mammal species.